# SET FUNCTIONS FOR TIME SERIES

## ABSTRACT

Despite the eminent successes of deep neural networks, many architectures are often hard to transfer to irregularly-sampled and asynchronous time series that occur in many real-world datasets, such as healthcare applications. This paper proposes a novel framework for classifying irregularly sampled time series with unaligned measurements, focusing on high scalability and data efficiency. Our method SEFT (**Set F**unctions for **T**ime Series) is based on recent advances in differentiable set function learning, extremely parallelizable, and scales well to very large datasets and online monitoring scenarios. We extensively compare our method to competitors on multiple healthcare time series datasets and show that it performs competitively whilst significantly reducing runtime.

## 1 INTRODUCTION

With the increasing digitalization, measurements over extensive time periods are becoming ubiquitous. Nevertheless, in many application domains, in particular healthcare (Yadav et al., 2018), measurements might not necessarily be observed at a regular rate or could be misaligned. Moreover, the presence or absence of a measurement and its observation frequency may carry information of its own (Little & Rubin, 2014), such that imputing the missing values is not always desired.

While some algorithms can be readily applied to datasets with varying length, these methods usually assume *regular* sampling of the data and/or require the measurements across modalities to be aligned/synchronized, preventing their application to the aforementioned settings. Existing approaches for *unaligned* measurements, by contrast, typically rely on imputation to obtain a regularly-sampled version of a dataset for classification. Learning a suitable imputation scheme, however, requires understanding the underlying dynamics of a system; this task is significantly more complicated and *not* necessarily required when classification is the main goal. Furthermore, even though a decoupled imputation scheme followed by classification is generally more scalable, it may lose information (in terms of "missingness patterns") that could be crucial for prediction tasks. In addition, the fact that decoupled schemes perform worse than methods that are trained end-to-end has been has been empirically demonstrated by Li & Marlin (2016). Approaches that jointly optimize both tasks also add a large computational overhead, thus suffering from poor scalability or high memory requirements.

Our method is motivated by the understanding that, while RNNs and similar architectures are well suited for capturing and modelling the dynamics of a time series and thus excel at tasks such as forecasting, retaining the order of an input sequence can even be a disadvantage in classification scenarios (Vinyals et al., 2015). We show that by relaxing the condition that a sequence must be processed in order, we can naturally derive an architecture that *directly* accounts for (i) irregular sampling, and (ii) unsynchronized measurements. Our method SEFT: **Set F**unctions for **T**ime Series, extends recent advances in set function learning to irregular sampled time series classification tasks, yields state-of-the-art performance, is highly scalable and improves over current approaches by almost an order of magnitude in terms of runtime.

With SEFT, we propose to rephrase the problem of classifying time series as classifying a set of observations. We show how *set functions* can be exploited to learn classifiers that are naturally applicable to unaligned and irregularly sampled time series, leading to state-of-the-art performance in irregularly-sampled time series classification tasks. Our approach can be interpreted as learning dataset-specific summary statistics of time series which are optimized to separate instances by class.

Furthermore, our method is highly parallelizable and can be readily extended to an online monitoring setup with up to thousands of patients.

## 2   RELATED WORK

This paper focuses on classifying time series with irregular sampling and potentially unaligned measurements. We briefly discuss recent work in this field; all approaches can be broadly grouped into the following three categories.

**Irregular sampling as missing data**   While the problem of supervised classification in the presence of missing data is closely related to irregular sampling on time series, there are some core differences. Missing data is usually defined with respect to a number of features that could be observed, whereas time series themselves can have different lengths and a "typical" number of observed values might not exist. Generally, an irregularly-sampled time series can be converted into a missing data problem by discretizing the time axis into non-overlapping intervals, and declaring intervals in which no data was sampled as missing. This approach is followed by Marlin et al. (2012), where a Gaussian Mixture Model was used to do semi-supervised clustering on electronic health records. Similarly, Lipton et al. (2016) discretize the time series into intervals, aggregate multiple measurements within an interval, and add missingness indicators to the input of a Recurrent Neural Network. By contrast, Che et al. (2018) present several variants of the Gated Recurrent Unit (GRU) combined with imputation schemes. Most prominently, the GRU-model was extended to include a decay term (GRU-D), such that the last observed value is decayed to the empirical mean of the time series via a learnable decay term. While these approaches are applicable to irregularly-sampled data, they either rely on imputation schemes or empirical global estimates on the data distribution (our method, by contrast, requires neither), without directly exploiting the global structure of the time series.

**Frameworks supporting irregular sampling**   Some frameworks support missing data. For example, Lu et al. (2008) directly defined a kernel on irregularly-sampled time series, permitting subsequent classification and regression with kernel-based classifiers or regression schemes. Furthermore, Gaussian Processes (Williams & Rasmussen, 2006) constitute a common probabilistic model for time series; they directly permit modelling of continuous time data using mean and covariance functions. Along these lines, Li & Marlin (2015) derived a kernel on Gaussian Process Posteriors, allowing the comparison and classification of irregularly-sampled time series using kernel-based classifiers. Nevertheless, all of these approaches still rely on separate tuning/training of the imputation method and the classifier so that structures supporting the classification could be potentially missed in the imputation step. An emerging line of research employs *Hawkes processes* (Hawkes, 1971; Liniger, 2009), i.e. a specific class of self-exciting point processes, for time series modelling and forecasting (Mei & Eisner, 2017; Yang et al., 2017; Xiao et al., 2017). While Hawkes processes exhibit extraordinary performance in these domains, there is no standardised way of using them for classification. Previous work (Lukasik et al., 2016) trains *multiple* Hawkes processes (one for each label) and classifies a time series by assigning it the label that maximises the respective likelihood function. Since this approach does not scale to our datasets, we were unable to perform a fair comparison. We conjecture that further research will be required to make Hawkes processes applicable to general time series classification scenarios.

**End-to-end learning of imputation schemes**   Methods of this type are composed of two modules with separate responsibilities, namely an imputation scheme and a classifier, where both components are trained discriminatively and end-to-end using gradient-based training. Recently, Li & Marlin (2016) proposed the Gaussian Process Adapters (GP Adapters) framework, where the parameters of a Gaussian Process Kernel are trained alongside a classifier. The Gaussian Process gives rise to a fixed-size representation of the irregularly-sampled time series, making it possible to apply *any* differentiable classification architecture. This approach was further extended to multivariate time series by Futoma et al. (2017) using Multi-task Gaussian Processes (MGPs) (Bonilla et al., 2008), which allow correlations between the imputed channels. Moreover, Futoma et al. (2017) made the approach more compatible with time series of different lengths by applying a Long Short Term Memory (LSTM) (Hochreiter & Schmidhuber, 1997) classifier. Motivated by the limited scalability of approaches based on GP Adapters, Shukla & Marlin (2019) suggest an alternative imputation

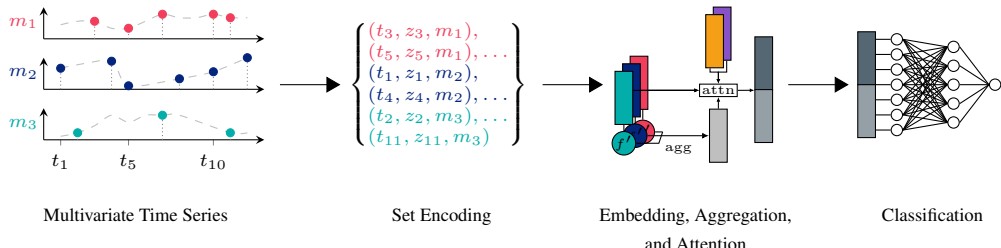

Figure 1: Schematic overview of SEFT's architecture. The first panel exemplifies a potential input, namely a multivariate time series, consisting of 3 modalities $m_1, m_2, m_3$. We treat the $j^{\text{th}}$ observation as a tuple $(t_j, z_j, m_j)$, comprising a time $t_j$, a value $z_j$, and a modality indicator $m_j$. All observations are summarized as a set of such tuples. Each set of tuples belonging to the same modality is then separately embedded ($f'$) and subsequently aggregated (agg). An attention mechanism (attn) as described in Section 3.3 is then applied to learn the importance of individual and consecutive observations. Respective query matrices for 2 attentions head are illustrated in purple and orange blocks. The results of each attention head are then concatenated and used as the input for final classification layers.

scheme, the *interpolation prediction networks*. It applies multiple semi-parametric interpolation schemes to obtain a regularly-sampled time series representation. The parameters of the interpolation network are trained with the classifier in an end-to-end setup.

## 3 PROPOSED METHOD

Our paper focuses on the problem of time series classification of irregularly sampled and unaligned time series. We first define the required terms before describing our models

### 3.1 NOTATION & REQUIREMENTS

**Definition 1** (Time series). *We describe a time series of an instance $i$ as a set $\mathcal{S}_i$ of $M := \text{len}(\mathcal{S}_i)$ observations $s_j$ such that $\mathcal{S}_i := \{s_1, \ldots, s_M\}$. We assume each observation $s_j$ to be represented as a tuple $(t_j, z_j, m_j)$, consisting of a time $t_j \in \mathbb{R}^+$, an observed value $z_j \in \mathbb{R}$, and a modality indicator $m_j \in \{1 \ldots D\}$, where $D$ represents the dimensionality of the time series. We write $\Omega \subseteq \mathbb{R}^+ \times \mathbb{R} \times \mathbb{N}^+$ to denote the domain of observations. An entire time series can thus be represented as*

$$\mathcal{S}_i := \{(t_1, z_1, m_1), \ldots, (t_M, z_M, m_M)\}, \tag{1}$$

*where for notational convenience we omitted the index $i$.*

We leave this definition very general on purpose, allowing the length of each time series (comprising all channels, such as "heart rate", "respiratory rate", etc. of one instance) to differ, since our models are capable of handling this. Likewise, we neither enforce nor expect all time series to be synchronized, i.e. being sampled at the same time, but rather we permit unaligned or *non-synchronized* observations in the sense of not having to observe all modalities at each time point. Time series are collected in a dataset $\mathcal{D}$.

**Definition 2** (Dataset). *We consider a dataset $\mathcal{D}$ to contain $n$ time series. Elements of $\mathcal{D}$ are tuples, i.e. $\mathcal{D} := \{(\mathcal{S}_1, y_1), \ldots, (\mathcal{S}_N, y_N)\}$, where $\mathcal{S}_i$ denotes the $i^{\text{th}}$ time series and $y_i \in \{1, \ldots, C\}$ its associated class label.*

Figure 1 gives a high-level overview of our method, including the individual steps required to perform classification. To get a more intuitive grasp of these definitions, we briefly illustrate our time series notation with an example. Let instance $i$ be an in-hospital patient, while the time series represent measurements of two channels of vital parameters during a hospital stay, namely heart rate (HR) and mean arterial blood pressure (MAP). We enumerate those channels as modalities 1

and 2. Counting from admission time, a HR of 60 and 65 beats per minute was measured after 0.5 h and 3.0 h, respectively, whereas MAP values of 80, 85, and 87 mmHg were observed after 0.5 h, 1.7 h, and 2.5 h. According to Definition 1, the time series is thus represented as $\mathcal{S}_i = \{(0.5, 60, 1), (3, 65, 1), (0.5, 80, 2), (1.7, 85, 2), (3, 87, 2)\}$. In this example, observations are ordered by modality to increase readability; in practice, we are dealing with unordered sets.

**Definition 3** (Non-synchronized time series). *We call a D-dimensional time series non-synchronized if there is at least one time point $t_j \in \mathbb{R}^+$ at which at least one modality is not observed, i.e. if there exists $t_j \in \mathbb{R}^+$ such that $|\{(t_k, z_k, m_k) \mid t_k = t_j\}| \neq D$.*

Furthermore, we assume that no two measurements of the same modality $m_k$ occur at the same time, i.e. $t_i \neq t_j$ for $i \neq j$ has to be satisfied for all measurements in $m_k$. This assumption is not required for *technical* reasons but for *consistency*. It also makes it possible to interpret the results later on.

To summarize our generic setup, we do not require $M$, the number of observations per time series, to be the same, i.e. $\text{len}(\mathcal{S}_i) \neq \text{len}(\mathcal{S}_j)$ for $i \neq j$ is permitted, nor do we assume that the time points and modalities of the observations are the same across time series. This setting is common in biomedical time series, for example. Since typical machine learning algorithms are designed to operate on data of a *fixed* dimension, novel approaches to this non-trivial problem are required.

## 3.2 OUR MODEL

In the following, we describe an approach inspired by differentiable learning of functions that operate on sets (Zaheer et al., 2017; Wagstaff et al., 2019). We phrase the problem of classifying time series on irregular grids as learning a function $f$ on a set of arbitrarily many time series observations following Definition 1, i.e. $\mathcal{S} = \{(t_1, z_1, m_1), \ldots, (t_M, z_M, m_M)\}$, such that $f \colon \mathcal{S} \to \mathbb{R}^C$, where $\mathcal{S}$ represents a generic time series of arbitrary cardinality and $\mathbb{R}^C$ corresponds to the logits of the $C$ classes in the dataset. As we previously discussed, we interpret each time series as an unordered set of measurements, where all information is conserved because the observation time is included for each set element. Specifically, we define $f$ to be a set function, i.e. a function that operates on a set and thus has to be *invariant* to the ordering of the elements in the set. Multiple architectures are applicable to constructing set functions such as Transformers (Lee et al., 2019; Vaswani et al., 2017), or Deep Sets (Zaheer et al., 2017). Due to preliminary experiments, where Transformers suffered from lower generalization performance in our setting[1], we base this work on the framework of Zaheer et al. (2017). Intuitively, this can be seen as computing multivariate dataset-specific summary statistics, which are optimized to maximize classification performance. Thus, we *sum-decompose* the set function $f$ into the form

$$f(\mathcal{S}) = g\left(\frac{1}{|\mathcal{S}|} \sum_{s_j \in \mathcal{S}} h(s_j)\right) \tag{2}$$

where $h \colon \Omega \to \mathbb{R}^d$ and $g \colon \mathbb{R}^d \to \mathbb{R}^C$ are neural networks, $d \in \mathbb{N}^+$ determines the dimensionality of the latent representation, and $s_j$ represents a single observation of the time series $\mathcal{S}$. We can view the averaged representations $1/|\mathcal{S}| \sum_{s_j \in \mathcal{S}} h(s_j)$ in general as a dataset-specific summary statistic learned to *best* distinguish the class labels. Equation 2 also implies the beneficial scalability properties of our approach: each embedding can be calculated independently of the others; hence, the constant computational cost of passing a single observation through the function $h$ is scaled by the number of observations, resulting in a runtime of $\mathcal{O}(M)$ for a time series of length $M$.

Recently, Wagstaff et al. (2019) derived requirements for a practical universal function representation of *sum-decomposable* set functions, i.e the requirements *necessary* for a *sum-decomposable* function to represent an arbitrary set-function given that $h$ and $g$ are arbitrarily expressive. In particular, they show that a universal function representation can only be guaranteed provided that $d \geq \max_i \text{len}(\mathcal{S}_i)$ is satisfied. During hyperparameter search we thus independently sample the dimensionality of the aggregation space, and allow it to be in the order of the number of observations that are to be expected in the dataset. Further, we explored the utilization of max, sum, and mean as alternative aggregation functions inspired by Zaheer et al. (2017); Garnelo et al. (2018).

---

[1]Please see Section 4.4 for a quantification.

**Intuition**  Our method can be connected to Takens's embedding theorem (Takens, 1981) for dynamical systems: we also observe a set of samples from some unknown (but deterministic) dynamical process; provided the dimensionality of our architecture is sufficiently large[2], we are capable of reconstructing the system up to diffeomorphism. The crucial difference is that we do *not* have to construct a time-delay embedding but rather, we let the network learn an embedding that is suitable for classification.

**Time encoding**  In order to represent the time point of an observation on a normalized scale, we employ variant of *positional encodings*, as introduced by Vaswani et al. (2017). Preliminary results indicated that this encoding scheme reduces the sensitivity towards initialization and training hyperparameters of a model. Specifically, the time encoding converts the one-dimensional time axis into a multi-dimensional input by passing the time $t$ of each observation through multiple sine and cosine functions of varying frequencies. Given a dimensionality $\tau \in \mathbb{N}^+$ of the time encoding, we refer to the encoded position as $x \in \mathbb{R}^\tau$, where

$$x_{2k}(t) := \sin\left(\frac{t}{\text{max\_ts}^{2k/\tau}}\right) \tag{3}$$

$$x_{2k+1}(t) := \cos\left(\frac{t}{\text{max\_ts}^{2k/\tau}}\right) \tag{4}$$

with $k \in \{0, \ldots, \tau/2\}$ and max_ts representing the maximal time scale that is expected in the data. Intuitively, we select the wavelengths using a geometric progression from $2\pi$ to $\text{max\_ts} \cdot 2\pi$, and treat the number of steps and the maximum timescale max_ts as hyperparameters of the model. For all experiments time encodings were used, such that an observation is represented as $s_j = (x(t_j), z_j, m_j)$.

**Loss function**  If not mentioned otherwise, we choose $h$ and $g$ in Equation 2 to be *multilayer perceptron* deep neural networks, parametrized by weights $\theta$ and $\psi$, respectively. We thus denote these neural networks by $h_\theta$ and $g_\psi$; their parameters are shared across all instances per dataset. In our training setup, we follow Zaheer et al. (2017) and apply the devised set function to the complete time series, i.e. to the set of all observations for each time series. Overall, we optimize a loss function that is defined as

$$\mathcal{L}(\theta, \psi) := \mathbb{E}_{(\mathcal{S},y)\in\mathcal{D}}\left[\ell\left(y; g_\psi\left(\frac{1}{|\mathcal{S}|}\sum_{s_j\in\mathcal{S}} h_\theta(s_j)\right)\right)\right], \tag{5}$$

where $\ell(\cdot)$ represents a task-specific loss function. In out setup, we either utilize the binary cross-entropy in combination with a sigmoid activation function in the last layer for binary classification or multi-label classification tasks and categorical cross-entropy in combination with a softmax activation function in the last layer for multi-class classification tasks.

### 3.3 Attention-based aggregation

So far, our method permits encoding sets of arbitrary sizes into a fixed-size representation. For increasingly large set sizes, however, many irrelevant observations could influence the result of the set function. The *mean* aggregation function is particularly susceptible to this because the influence of an observation to the embedding shrinks proportionally to the size of the set. We thus suggest to use a *weighted* mean in order to allow the model to decide which observations are relevant and which should be considered irrelevant. This is equivalent to computing an attention $a(\mathcal{S}, s_j)$ over the set input elements, and subsequently, computing the sum over all elements in the set.

Our approach is based on *scaled dot-product attention* with multiple heads $i \in \{1, \ldots, m\}$ in order to be able to cover different aspects of the aggregated set[3]. We define $a(\cdot)$, i.e. the attention weight function of an individual time series, to depend on the overall set of observations. This is achieved

---

[2]In Takens's embedding theorem, $d > d_B$ is required, where $d_B$ refers to the fractal box counting dimension (Liebovitch & Toth, 1989), which is typically well below the size of typical neural network architectures.

[3]Since we are dealing only with a single instance (i.e. time series) in this section, we use $i$ and $j$ to denote a *head* and an *observation*, respectively.

by computing an embedding of the set elements using a smaller set function $f'$, and projecting the concatenation of the set representation and the individual set elements into a $d$-dimensional space. Specifically, we have $K_{j,i} = [f'(\mathcal{S}), s_j]^T W_i$ where $W_i \in \mathbb{R}^{(\text{im}(f')+|s_j|) \times d}$ and $K \in \mathbb{R}^{|\mathcal{S}| \times d}$. Furthermore, we define a matrix of query points $Q \in \mathbb{R}^{m \times d}$, which allow the model to summarize different aspects of the dataset via

$$e_{j,i} = \frac{K_{j,i} \cdot Q_i}{\sqrt{d}} \qquad \text{and} \qquad a_{j,i} = \frac{\exp(e_{j,i})}{\sum_j \exp(e_{j,i})}$$

where $a_{j,i}$ represents the amount of attention that head $i$ gives to set element $j$. The head-specific row $Q_i$ of the query matrix $Q$ allows a head to focus on individual aspects (such as the distribution of one or multiple modalities) of a time series. For each head, we multiply the set element embeddings computed via the set function $f$ with the attentions derived for the individual instances, i.e. $r_i = \sum_j a_{j,i} f(s_j)$. The computed representation is concatenated and passed to the aggregation network $h_\theta$ as in a regular set function, i.e. $r* = [r_1 \dots r_m]$. In our setup, we initialize $Q$ with zeros, such that at the beginning of training, the attention mechanism is equivalent to computing the unweighted mean over the set elements.

Overall, this aggregation function is similar to Transformers (Vaswani et al., 2017), but differs from them in a few key aspects. Standard Transformer blocks would *use the information from all set elements* in order to compute the embedding of an individual set element, leading to a runtime and space complexity of $\mathcal{O}(n^2)$. In contrast, our approach computes the embeddings of set elements independently, leading lower runtime and memory complexity of $\mathcal{O}(n)$. Further, we observed that computing embeddings with information from other set elements (as the Transformer does) actually *decreases generalization performance* (see Table 1 for details).

## 4 EXPERIMENTS

We executed all experiments and implementations in a unified code base, which we also make available[4] to the community. While some of the datasets used subsequently have access restrictions, anybody can gain access after satisfying the defined requirements. This ensures the reproducibility of our results. Please consult Appendix A.2 for further details.

### 4.1 DATASETS

In order to benchmark the proposed method we selected 4 datasets with irregularly-sampled and non-synchronized measurements.

**Healing MNIST** The `H-MNIST` dataset was introduced by Krishnan et al. (2015) in order to simulate characteristics which typically occur in medical time series. In our setup, we use a variant of this dataset. Every instance of the dataset contains 10 frames, derived from a single instance of `MNIST` dataset, where the digit is rotated according to an angle uniformly sampled between $-90°$ to $90°$. Furthermore, 3 randomly-selected consecutive frames are augmented by a square artefact in the top left corner of the image in order to indicate seasonality in the time series. Finally, 60 % of the data points are randomly discarded in order to yield a final high-dimensional irregularly-sampled time series with non-synchronized measurements. Using these settings each instance has on average $3,136$ observations.

**MIMIC-III Tasks** MIMIC-III (Johnson et al., 2016) is a widely-used, freely-accessible dataset containing around $50,000$ distinct ICU stays. The median length of stay is 2.1 d and a wide range of physiological measurements (e.g. arterial blood pressure, respiration rate, heart rate) are recorded with a resolution of 1 h. Furthermore, laboratory test results, collected at irregular time intervals are available. Recently, Harutyunyan et al. (2019) defined a set of machine learning tasks, labels, and benchmarks using a subset of the MIMIC-III dataset. We trained and evaluated our method and competing methods on the binary mortality prediction task (`M3-Mortality`) and on the multi-class problem of phenotype classification (`M3-Phenotyping`), while applying additional filtering described in Appendix A.1. The goal of the mortality prediction task is to predict whether a patient will die during his/her hospital stay using only data from the first 48 hours of the ICU stay. This

---

[4]`https://osf.io/2hg74/?view_only=8d45fdf237954948a02f1e2bf701cdf1`

dataset contains around $21,000$ stays of which approximately $10\%$ result in death. The phenotype classification task consists of $40,000$ patients, each of which can suffer from a multitude of 25 acute care conditions.

**Physionet Mortality Prediction Challenge** The 2012 Physionet challenge dataset (Goldberger et al., 2000), which we abbreviate `P-Mortality`, contains $12,000$ ICU stays each of which lasts at least 48 h. For each stay, a set of general descriptors (such as gender, age, height, weight) were collected at admission time. Depending on the course of the stay and patient status, up to 37 time series variables were measured (e.g. blood pressure, lactate, respiration rate, temperature). While some modalities might be measured in regular time intervals (e.g. hourly or daily), some are only collected when required. Not all variables are available for each stay. The goal of the challenge was to predict if—and with which certainty —a patient will die during the hospital stay. The training set consists of $8,000$ stays while the testing set comprises $4,000$ ICU visits. Both datasets are similarly imbalanced, with a prevalence of around $14\%$. For simplicity, the general descriptors (such as age and weight), were included as time points with a single observation at the beginning of the stay. This treatment is similar to the approach by Harutyunyan et al. (2019) in the MIMIC-III benchmarking datasets. Please refer to Table A.1, Table A.2, and Table A.3 in the appendix for a more detailed enumeration of samples sizes and label distributions. The total number of samples may slightly deviate from the originally published splits, as time series of excessive length prevented fitting some methods in reasonable time, and were therefore excluded.

## 4.2 COMPETITOR METHODS

**GRU-simple** GRU-SIMPLE (Che et al., 2018) augments the input at time $t$ of a Gated-Recurrent-Unit RNN with a measurement mask $m_t^d$ and a $\delta_t$ matrix, which contains the time since the last measurement of the corresponding modality $d$, such that

$$\delta_t = \begin{cases} s_t - s_{t-1} + \delta_{t-1}^d & t > 1, m_{t-1}^d = 0 \\ s_t - s_{t-1} & t > 1, m_{t-1}^d = 1 \\ 0 & t = 0 \end{cases}$$

where $s_t$ represents the time associated with time step $t$.

**Phased-LSTM** The PHASED-LSTM (Neil et al., 2016) introduced a biologically inspired time dependent gating mechanism which regulates access to the hidden and cell state of a Long short-term RNN cell (Hochreiter & Schmidhuber, 1997). While this allows the network to handle event-based sequences with irregularly spaced observations, the approach does not support unaligned measurements. In order to still provide the architecture with all relevant information, we augment the input in a similar fashion as described for the GRU-SIMPLE approach.

**GRU-D** GRU-D or GRU-Decay (Che et al., 2018) contains modifications to the GRU RNN cell, allowing it to decay past observations to the mean imputation of a modality using a learnable decay rate. By additionally providing the measurement masks as an input the recurrent neural network the last feed in value. Learns how fast to decay back to a mean imputation of the missing data modality.

**Interpolation Prediction Networks** IP-NETWORKS (Shukla & Marlin, 2019) apply multiple semi-parametric interpolation schemes to irregularly-sampled time series to obtain regularly-sampled representations that cover long-term trends, transients, and also sampling information. The method combines a univariate interpolation step with a subsequent multivariate interpolation; the parameters of the interpolation network are trained with the classifier in an end-to-end fashion.

**Transformer** In the TRANSFORMER architecture (Vaswani et al., 2017) the elements of a sequence are encoded simultaneously and information between sequence elements is captured using Multi-Head-Attention blocks. In our case, an individual sequence element corresponds to all measurements available at a given time point, augmented with a measurement indicator. Transformers are normally used for sequence-to-sequence modelling tasks and in our setup were adapted to classification tasks by mean-aggregating the final representation. This representation is then fed into a one-layer MLP to predict logits for the individual classes.

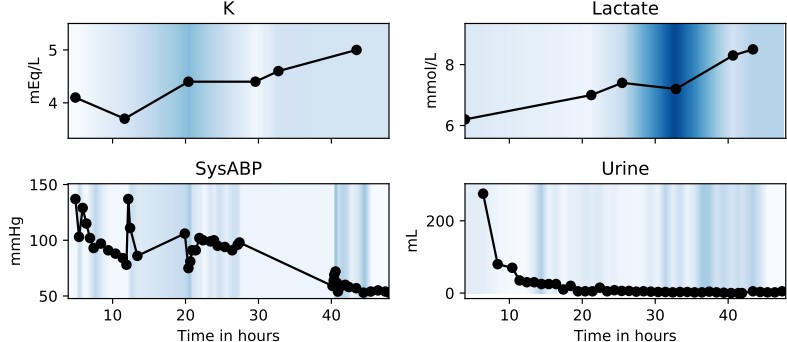

Figure 2: Visualizations of a single attention head on an instance of the `P-Mortality` dataset. We display a set of variables relevant for assessing patient stability and organ failure: Serum Potassium (K), Lactate, Systolic Arterial Blood Pressure (SysABP), and Urine output. Darker colors represent higher attention values.

### 4.3 EXPERIMENTAL SETUP

To permit a fair comparison between the methods, we executed hyperparameter searches for each model on each dataset, composed of uniformly sampling 20 parameters according to Appendix A.3. Training was stopped after 20 epochs without improvement of the validation loss, the hyperparameters with the best overall validation performance were selected for quantifying the performance on the test set. The train, validation, and test splits were the same for all models and all evaluations. Final performance on the test set was calculated by 3 *independent* runs of the models; evaluation took place after the model was restored to the state with the best validation loss. In all subsequent benchmarks, we use the standard deviation of the test performance of these runs as generalization performance estimates.

### 4.4 RESULTS

The results are shown in Table 1. Overall, our proposed method exhibits the *lowest* per-epoch runtime on most datasets, while either yielding competitive or state-the-art performance. Further, the trade-off between runtime and performance of the proposed method is very good on all datasets (see Figure A.1 and Figure A.2 in the appendix for a visualization of this argument). In order to elucidate the contribution of individual model components, we also provide an ablation study in Table A.4. Here we see that the attention mechanism contributes more to the model performance, while the positional encoding seems to be beneficial for datasets with highly-varying time series lengths, in particular `M3-Phenotyping`.

**Opening the black box** In the medical domain, it is of particular interest to understand the decisions a model makes based on the input it is provided with. The formulation of our model and its *per observation* perspective on time series gives it the unique property of being able to quantify to which extent an individual observation contributed to the output of the model. We exemplify this in Figure 2 with a patient time series that was combined with our models attention values, displayed for a set of clinically relevant variables. After reviewing these records with our medical expert, we find that our model is able to pick up regions with drastic changes in individual modalities. Moreover, it is able to inspect other modalities at the same associated time (for instance, at hour 20). This is behaviour similar to what one would expect from an alerted clinician reviewing the logged medical records. Interestingly, we observe that the model attends to *known* trends (that are consisting with domain knowledge about patient deterioration ultimately resulting in death) such as increase in lactate or hemodynamic instability, as indicated by drops in blood pressure. Furthermore, the model appears to be alerted by persisting low urine output. After several hours, this can be indicative of kidney failure.

Table 1: Performance comparison of methods on benchmarking datasets. Performance metrics have been rescaled to 100 for readability reasons. "AUC" denotes the area under the Receiver Operating Characteristic (ROC) curve; "PR AUC" denotes the area under the precision recall curve. "MICRO" refers to evaluating the metric globally by treating each entry of the label indicator matrix as a label. For "MACRO", the metric is computed for each class and then averaged, whereas in "WEIGHTED" the class-wise metrics are weighted by class imbalance. Values denoted with "OOM" were not obtainable due to restrictions in GPU memory.

| DATASET | MODEL | MICRO AUC | MACRO AUC | WEIGHTED AUC | RUNTIME |
|---|---|---|---|---|---|
| H-MNIST | GRU-SIMPLE | $99.09 \pm 0.05$ | $99.01 \pm 0.05$ | $99.03 \pm 0.05$ | $11.43 \pm 0.47$ |
| | PHASED-LSTM | $98.63 \pm 0.13$ | $98.50 \pm 0.15$ | $98.52 \pm 0.14$ | $33.93 \pm 1.11$ |
| | GRU-D | $99.42 \pm 0.01$ | $99.37 \pm 0.02$ | $99.38 \pm 0.02$ | $\mathbf{11.81 \pm 0.44}$ |
| | IP-NETS | $99.06 \pm 0.05$ | $98.96 \pm 0.03$ | $98.98 \pm 0.03$ | $127.76 \pm 0.95$ |
| | TRANSFORMER | $\mathbf{99.59 \pm 0.05}$ | $\mathbf{99.55 \pm 0.05}$ | $\mathbf{99.56 \pm 0.05}$ | $21.62 \pm 0.90$ |
| | SEFT* | $\underline{\mathbf{99.76 \pm 0.01}}$ | $\underline{\mathbf{99.75 \pm 0.01}}$ | $\underline{\mathbf{99.75 \pm 0.01}}$ | $\underline{\mathbf{4.05 \pm 0.35}}$ |
| M3-Phenotyping | GRU-SIMPLE | $79.89 \pm 0.14$ | $73.91 \pm 0.19$ | $72.55 \pm 0.16$ | $112.58 \pm 2.03$ |
| | PHASED-LSTM | $80.00 \pm 0.06$ | $73.91 \pm 0.09$ | $72.65 \pm 0.08$ | $400.41 \pm 14.14$ |
| | GRU-D | $\underline{\mathbf{82.16 \pm 0.04}}$ | $\underline{\mathbf{77.14 \pm 0.03}}$ | $\underline{\mathbf{76.08 \pm 0.01}}$ | $288.70 \pm 16.66$ |
| | IP-NETS | —OOM— | —OOM— | —OOM— | —OOM— |
| | TRANSFORMER | —OOM— | —OOM— | —OOM— | —OOM— |
| | SEFT | $81.22 \pm 0.12$ | $75.95 \pm 0.09$ | $74.90 \pm 0.11$ | $\mathbf{56.27 \pm 2.14}$ |
| | SEFT-ATTN | $\mathbf{82.00 \pm 0.06}$ | $\mathbf{76.95 \pm 0.09}$ | $\mathbf{75.88 \pm 0.09}$ | $\underline{\mathbf{52.32 \pm 0.74}}$ |
| | | ACCURACY | PR AUC | AUC | RUNTIME |
| M3-Mortality | GRU-SIMPLE | $88.24 \pm 0.38$ | $36.36 \pm 1.31$ | $79.36 \pm 0.26$ | $22.80 \pm 0.56$ |
| | PHASED-LSTM | $88.32 \pm 0.31$ | $35.30 \pm 1.38$ | $80.16 \pm 0.22$ | $25.54 \pm 0.26$ |
| | GRU-D | $\mathbf{89.56 \pm 0.38}$ | $\underline{\mathbf{46.76 \pm 0.65}}$ | $\mathbf{83.73 \pm 0.21}$ | $31.85 \pm 0.86$ |
| | IP-NETS | $\underline{\mathbf{89.73 \pm 0.16}}$ | $\mathbf{45.88 \pm 0.87}$ | $83.30 \pm 0.56$ | $101.12 \pm 4.52$ |
| | TRANSFORMER | $89.14 \pm 0.15$ | $42.32 \pm 0.41$ | $82.60 \pm 0.55$ | $\mathbf{4.79 \pm 0.02}$ |
| | SEFT | $88.65 \pm 0.49$ | $36.18 \pm 5.07$ | $79.15 \pm 3.00$ | $\underline{\mathbf{3.72 \pm 0.11}}$ |
| | SEFT-ATTN | $89.48 \pm 0.16$ | $45.25 \pm 0.96$ | $\underline{\mathbf{83.79 \pm 0.59}}$ | $16.64 \pm 0.20$ |
| P-Mortality | GRU-SIMPLE | $85.66 \pm 0.14$ | $39.43 \pm 0.71$ | $79.79 \pm 0.16$ | $5.16 \pm 0.06$ |
| | PHASED-LSTM | $85.57 \pm 0.11$ | $39.55 \pm 0.62$ | $78.71 \pm 0.76$ | $18.59 \pm 1.15$ |
| | GRU-D | $87.19 \pm 0.30$ | $\underline{\mathbf{54.95 \pm 0.54}}$ | $\underline{\mathbf{86.58 \pm 0.32}}$ | $14.08 \pm 0.38$ |
| | IP-NETS | $\mathbf{87.23 \pm 0.18}$ | $\mathbf{54.87 \pm 0.41}$ | $\mathbf{86.42 \pm 0.18}$ | $7.21 \pm 0.46$ |
| | TRANSFORMER | $86.47 \pm 0.08$ | $48.72 \pm 0.61$ | $83.49 \pm 0.46$ | $\underline{\mathbf{2.69 \pm 0.43}}$ |
| | SEFT | $87.11 \pm 0.32$ | $52.07 \pm 0.41$ | $84.12 \pm 0.32$ | $\mathbf{3.07 \pm 0.03}$ |
| | SEFT-ATTN | $\underline{\mathbf{87.62 \pm 0.16}}$ | $54.05 \pm 0.27$ | $85.50 \pm 0.13$ | $7.54 \pm 0.08$ |

*: Due to the high dimensionality of H-MNIST and associated memory issues, the set elements were constructed by concatenating the observation time with all values associated with the time point and measurement indicators. Furthermore, as this dataset features only 10 time steps and missingness is induced randomly, we refrained from applying the attention-based aggregation.

## 5 CONCLUSION

In this work, we presented a novel approach for classifying time series with irregularly-sampled and unaligned, that is *non-synchronized*, observations. Our approach yields state-of-the-art to strongly competitive performance on numerous simulated and real-world datasets, while reducing runtime by almost half. Moreover, we demonstrated that combining the perspective of individual observations with an attention mechanism permits increasing the interpretability of the model. This is particularly relevant for the medical and healthcare applications.

For future work, we reserve a more extensive exploration of the learned latent representation to evaluate its utility for clustering of time series or visualization of their similarity.

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

# A APPENDIX

Table A.1: `M3-Mortality` prevalence of labels for the binary classification task

|  | Training Prevalence | Testing Prevalence |
|---|---|---|
| In-hospital deaths | 0.135 | 0.116 |

Table A.2: `P-Mortality` prevalence of labels for the binary classification task

|  | Training Prevalence | Testing Prevalence |
|---|---|---|
| In-hospital deaths | 0.140 | 0.146 |

## A.1 DATA FILTERING

Due to memory requirements of some of the competitor methods, it was necassary to excluded time series with extremly high number of measurements. For the `M3-Phenotyping` patients with more than 2000 distinct time points were discarded from training. For `M3-Mortality` patients with more than 1000 time points were discarded as they contained dramatically different measuring frequencies compared to the rest of the dataset.

## A.2 IMPLEMENTATIONAL DETAILS

All experiments were run using `tensorflow 0.15.0rc0` and training was performed on `NVIDIA Geforce GTX 1080` GPUs. In order to allow a fair comparison between methods, the input processing pipeline cached model specific representations and transformations of the data. To further increase efficiency of the RNNs, sequences were binned in to buckets of jointly trained instances depending on their sequence length. The buckets were determined according to the $(0.25, 0.5, 0.75)$ quantiles of the length distributions of the datasets.

## A.3 TRAINING, MODEL ARCHITECTURES AND HYPERPARAMETER SEARCH

**General** All models were trained using the Adam optimizer, while randomly sampling the learning rate from $(0.001, 0.0005, 0.00025, 0.0001)$. Further, the batch size of all methods was sampled from the values $(32, 64, 128, 256)$.

**Recurrent neural networks** For the RNN based methods (GRU-SIMPLE, PHASED-LSTM, GRU-D and IP-NETS), the number of units was sampled in from the values $(16, 32, 64, 128, 256, 512)$. Further, recurrent dropout and input dropout were sampled from the values $(0.0, 0.1, 0.2, 0.3)$. Solely, for the PHASED-LSTM method, we did not apply dropout to the recurrent state and the inputs, as the learnt frequencies were hypothesized to fulfill a similar function as dropout (Neil et al., 2016).

**SEFT** We vary the number of layers, dropout in between the layers and the number of nodes per layer for both the encoding network $h_\theta$ and the aggregation network $g_\psi$ from the same ranges. The number of layers is randomly sampled between 1 and 5, the number of nodes in a layer are uniformly sampled from the range $(16, 32, 64, 128, 256, 512)$ and the dropout fraction is sampled from the values $(0.0, 0.1, 0.2, 0.3)$. The width of the embedding space prior to aggregation is sampled from the values $(32, 64, 128, 256, 512, 1024, 2048)$. The aggregation function selected to be one of $mean$, $sum$ and $max$. The number of dimensions used for the positional embedding $\tau$ is selected uniformly from $(4, 8, 16)$ and max_ts us selected from the values $(10, 100, 1000)$.

**SEFT-Attn** The parameters for the encoding and aggregation networks are sampled in a similar fashion as for SEFT. In contrast we set the aggregation function to be $sum$ as described in the text.

Table A.3: `M3-Phenotyping` prevalence of labels for the multi label classification task

| Phenotype | Training | Validation | Testing |
|---|---|---|---|
| Acute and unspecified renal failure | 0.216 | 0.207 | 0.211 |
| Acute cerebrovascular disease | 0.0746 | 0.0753 | 0.0662 |
| Acute myocardial infarction | 0.103 | 0.103 | 0.108 |
| Cardiac dysrhythmias | 0.322 | 0.317 | 0.323 |
| Chronic kidney disease | 0.135 | 0.131 | 0.132 |
| Chronic obstructive pulmonary disease and bronchiectasis | 0.132 | 0.128 | 0.126 |
| Complications of surgical procedures or medical care | 0.207 | 0.201 | 0.213 |
| Conduction disorders | 0.0726 | 0.07 | 0.0704 |
| Congestive heart failure; nonhypertensive | 0.268 | 0.264 | 0.268 |
| Coronary atherosclerosis and other heart disease | 0.323 | 0.317 | 0.331 |
| Diabetes mellitus with complications | 0.0955 | 0.0945 | 0.094 |
| Diabetes mellitus without complication | 0.194 | 0.187 | 0.192 |
| Disorders of lipid metabolism | 0.291 | 0.287 | 0.289 |
| Essential hypertension | 0.421 | 0.41 | 0.424 |
| Fluid and electrolyte disorders | 0.267 | 0.276 | 0.265 |
| Gastrointestinal hemorrhage | 0.0715 | 0.0747 | 0.0788 |
| Hypertension with complications and secondary hypertension | 0.133 | 0.131 | 0.13 |
| Other liver diseases | 0.0884 | 0.0904 | 0.0883 |
| Other lower respiratory disease | 0.0514 | 0.0484 | 0.0565 |
| Other upper respiratory disease | 0.0408 | 0.0371 | 0.0429 |
| Pleurisy; pneumothorax; pulmonary collapse | 0.0858 | 0.09 | 0.0905 |
| Pneumonia (except that caused by tuberculosis or sexually transmitted disease) | 0.14 | 0.135 | 0.135 |
| Respiratory failure; insufficiency; arrest (adult) | 0.18 | 0.184 | 0.177 |
| Septicemia (except in labor) | 0.142 | 0.145 | 0.138 |
| Shock | 0.0783 | 0.0745 | 0.0811 |
| Total samples | 29 208 | 6359 | 6266 |

Further we use a constant architecture for the attention network $f'$ with 2 layers, 64 nodes per layer, 4 heads and a dimensionality of the dot product space $d$ of 128. We solely sample the amount of attention dropout uniformly from the values $(0.0, 0.1, 0.25, 0.5)$.

**Transformer**   We utilize the same model architecture as defined in Vaswani et al. (2017), where we use a one hidden layer MLP as a feed-forward network, with dimensionality of the hidden layer selected to be twice the model dimensionality. The parameters for the Transformer network were sampled according to the following criteria. The dimensionality of the model was sampled uniformly from the values $(64, 128, 256, 512, 1024)$, the number of attention heads per layer from the values $(2, 4, 8)$ and the number of layers from the range $[1, 6] \in \mathbb{N}$. Further, we sampled the amount of dropout of the residual connections and the amount of attention dropout uniformly from the values $(0.0, 0.1, 0.2, 0.3, 0.5)$, and the maximal timescale for the time embedding from the values $(10, 100, 1000)$ (similar to the SEFT approach).

Table A.4: Ablation study of individual components of SEFT. "AUC" denotes the area under the Receiver Operating Characteristic (ROC) curve; "PR AUC" denotes the area under the precision recall curve; "RUNTIME" denotes the runtime of one training epoch in seconds.

| DATASET | MODEL | MICRO AUC | MACRO AUC | WEIGHTED AUC | RUNTIME |
|---|---|---|---|---|---|
| H-MNIST | SEFT | **99.76 ± 0.01** | **99.75 ± 0.01** | **99.75 ± 0.01** | 4.05 ± 0.35 |
| M3-Phenotyping | SEFT (NO ATTENTION) | 81.22 ± 0.12 | 75.95 ± 0.09 | 74.90 ± 0.11 | 56.27 ± 2.14 |
| | SEFT-ATTN (NO TIME ENC.) | 80.46 ± 0.86 | 74.70 ± 1.12 | 73.48 ± 1.18 | **50.17 ± 0.84** |
| | SEFT-ATTN | **82.00 ± 0.06** | **76.95 ± 0.09** | **75.88 ± 0.09** | 52.32 ± 0.74 |
| | | ACCURACY | PR AUC | AUC | RUNTIME |
| M3-Mortality | SEFT (NO ATTENTION) | 88.65 ± 0.49 | 36.18 ± 5.07 | 79.15 ± 3.00 | **3.72 ± 0.11** |
| | SEFT-ATTN (NO TIME ENC.) | 89.31 ± 0.08 | 44.12 ± 0.06 | 83.72 ± 0.34 | 17.60 ± 0.43 |
| | SEFT-ATTN | **89.48 ± 0.16** | **45.25 ± 0.96** | **83.79 ± 0.59** | 16.64 ± 0.20 |
| P-Mortality | SEFT (NO ATTENTION) | 87.11 ± 0.32 | 52.07 ± 0.41 | 84.12 ± 0.32 | **3.07 ± 0.03** |
| | SEFT-ATTN (NO TIME ENC.) | 87.03 ± 0.06 | 51.86 ± 1.04 | 84.91 ± 0.29 | 7.04 ± 0.04 |
| | SEFT-ATTN | **87.62 ± 0.16** | **54.05 ± 0.27** | **85.50 ± 0.13** | 7.54 ± 0.08 |

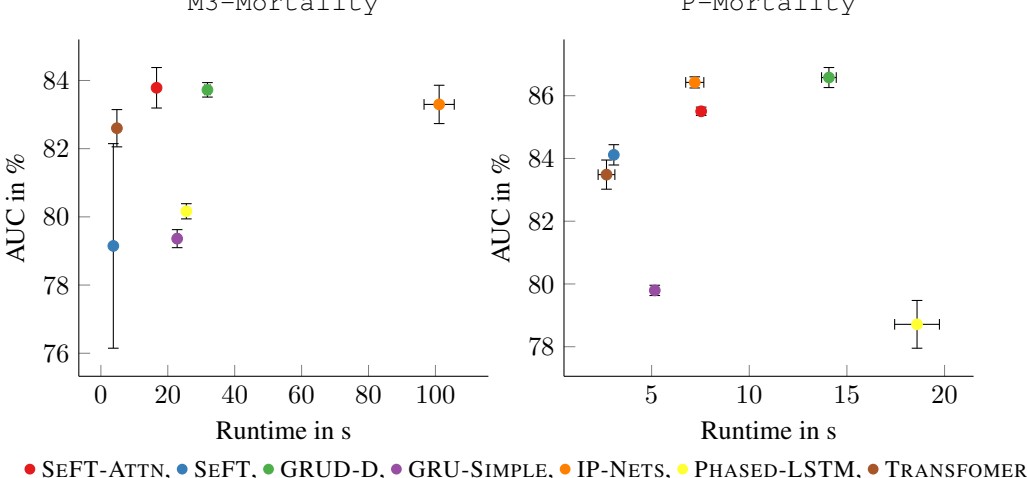

Figure A.1: A visualisation of the runtime of all methods and their AUC for datasets with a binary classification scenario.

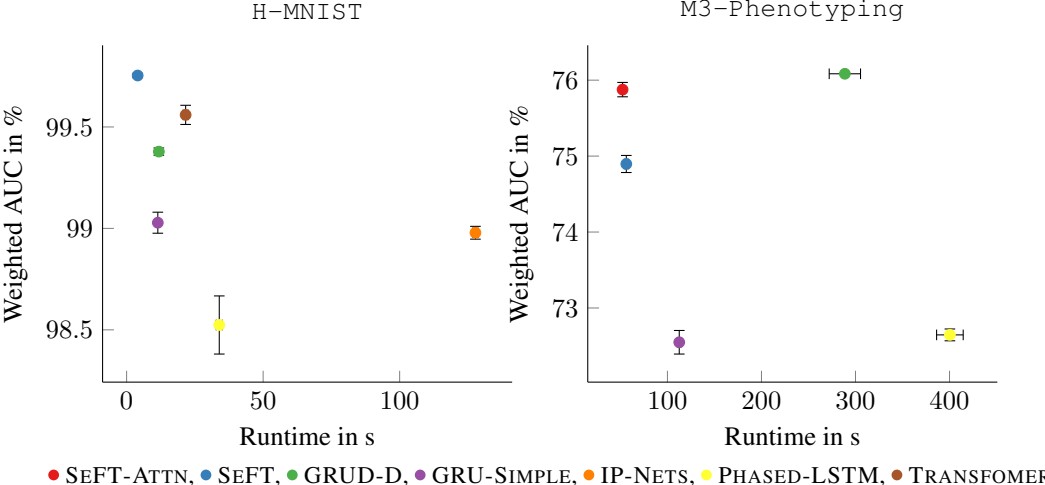

Figure A.2: A visualisation of the runtime of all methods and their AUC for datasets with a multilabel classification scenario. Please note that the model definition for SEFT changes between the left and the right column; please see Table 1 for more details.

