# OpenReview forum: "Set Functions for Time Series"
_ICLR.cc/2020/Conference — Reject_

### Official Review · AnonReviewer1 · 2019-10-21
**Official Blind Review #1**

**Rating:** 3

**Review:**

The paper idea of this paper is straightforward and clear: treat the irregular time series as a bag of events, augment them with time information using positional encoding, and process the events in parallel. The idea is certainly faster than the sequential algorithms such as RNNs and their extensions. However, as shown in the experiments, because it does not encode the "sequential-ness prior" to the model, it is less accurate. Compared to RNNs, the proposed model has better access to the entire length of sequences and does not suffer from the limited memory issues of RNNs and variants.

The proposed idea in this paper can be considered a simplified version of the Transformers. Like transformers, the time and order are only provided to the model using the positional encoding and attention is central to aggregation over the sequence. Realizing the relationship with the Transformers not only decreases the novelty degree for this paper but also requires the authors to include the Transformers in the baselines.

Finally, the results reported in the experiments are nice, especially for the baseline GRU-D! However, the MIMIC-III Mortality benchmark has a lot more than 21,000 stays to the best of my recollection. Can you please elaborate on how the number of data points has decreased?

**Experience Assessment:**

I have published one or two papers in this area.

**Review Assessment: Checking Correctness Of Derivations And Theory:**

I carefully checked the derivations and theory.

**Review Assessment: Checking Correctness Of Experiments:**

I assessed the sensibility of the experiments.

**Review Assessment: Thoroughness In Paper Reading:**

I read the paper thoroughly.

---

> ### Author Response · Authors · 2019-11-11
> **Our comments to your review**
>
> > Sequentialness prior
>
> Our approach is exhibiting competitive performance with respect to standard measures. To better showcase this, we added a visualisation (Figures A.1 and A.2 in the appendix) depicting the trade-offs in performance and runtime.
>
> > The proposed idea in this paper can be considered a simplified version of the Transformers architecture [...]
>
> Interestingly, in the initial phases of the project we found that the naive application of transformer architectures leads to low generalization performance and overfitting, despite the use of regularisation. We found that the independent encoding of individual observations *improves* the generalization performance. Hence, in contrast to transformers, we do not include interactions between set elements during the computation of the embedding, as this strategy regularises the network. This is in line with the observation that for medical time series simple machine learning methods with summary statistics can already achieve very good performance (see Harutyunyan et al. (2019), for more details). Using an independent encoding of the individual set elements thus corresponds to computing multivariate data set specific summary statistics which optimize the classification performance.
>
> We completely agree that including transformers as an additional baseline makes a lot of sense; we ran new experiments and updated the paper accordingly. In general, we observe that the performance of the transformer architecture exhibits competitive runtimes, but non-competitive classification performance. It was thus necessary to change parts of the results and discussion section to maintain a consistent story of the paper.
>
> > However, the MIMIC-III Mortality benchmark [...]
>
> In order to stay comparable with other research in the field, we decided to use the same dataset as Harutyunyan et al. (2019) and only remove the imputation and resampling steps to obtain a more raw, irregularly sampled version of the dataset. The original publication by Harutyunyan et al. (2019) consists of 21,139 patients (in their paper, see Page 10, Fig. 7). We discard 32 of these patients as they showed a drastically higher sampling rate, which lead to memory issues in the baselines. For further details, please refer to Section A.1 in the appendix.
>
> Please let us know if there are any other questions!

---

> > ### Comment · AnonReviewer1 · 2019-11-13
> > **Relationship with Transformers**
> >
> > Thank you for clarification on the relationship between this work and the Transformers and including them in the experiments. Transformers also embed the observations individually. Would you please explicitly mention in the paper that the Transformers are indeed "set functions" too?
> >
> > Can you provide further details on the structure of the Transformers that you have used as the baseline?

---

> > > ### Author Response · Authors · 2019-11-14
> > > **Further updates to the paper**
> > >
> > > Thank you for the further pointers. We now explicitly mention that Transformers are also set functions and refer to literature where they have been used as such.
> > >
> > > > Transformers also embed the observations individually.
> > >
> > > We totally agree, the key difference is that the embedding of the transformer also incorporates information from other elements being encoded.
> > >
> > > > Further details on the structure of Transformers used
> > >
> > > We now explicitly refer to the Transformer model architecture used for the experiments and add exact details in the section “Training, Model Architectures and Hyperparameter Search” of the appendix.
> > >
> > > Please let us know if there are any further questions!

---

### Official Review · AnonReviewer3 · 2019-10-23
**Official Blind Review #3**

**Rating:** 6

**Review:**

Summary:
The work is focused on classification of irregularly sampled and unaligned multi-modal time series. Prior work has primarily focused on imputation methods, either end-to-end or otherwise. This paper approaches the problem as a set function mapping between the time-series tuples to the class label. The proposed method is uses a set encoding of a multi-modal time series input, followed by mode-specific encoding of the tuples which are then aggregated in multiple ways prior to classification. An attention mechanism is attached in order for the model to automatically weigh the relevance of tuples for the classification. The model is compared to imputation based baselines on clinical ICU time series classification tasks. The performance mostly appears comparable across baselines but the proposed method has much better run-times.

The paper is for the most part well written, and related work well characterized. The formulation is interesting and clinically relevant as well so the choice of data-sets makes some sense. I have a few concerns about the architecture formulation and lack of clarification and intuition in what appears to be the main contribution of the paper (Sec 3.2 and 3.3) which I will detail below:

a. In the evaluation, I really want to see a decoupling between the "time encoding step" and "attention based aggregation" on the performance to figure out to isolate different sources of performance improvements. That is can there be a SEFT without time encoding? If not, why not? I encourage more ablation like studies that look at different sources of performance gains and demonstrate them in experiments.

b. The description of Sec 3.3. is really missing key motivation for the choices made around how the attention formulation is designed. For example why does the dot produce include the set elements? What if it doesn't? What is Q supposed to capture?

c. Is a_{j,i} shared across instances? Then irrespective of the number of observations per instance, the $j^{th}$ tuple gets similar weights? If not appropriate indexing will help clarify this.

d. It would be useful to provide how exactly a label is inferred for a *new* test instance.

I have some minor additional feedback (just for presentation and motivation purposes):

1. Authors make a claim in the introduction which should likely be qualified with a citation - "Furthermore, even though a decoupled imputation scheme followed by classification is generally more scalable, it may lose information that is relevant for prediction tasks". How does decoupled imputation imply loss of relevant information? By losing information about which observations are missing and relying on that for prediction? Does this clinically make sense? Or even generally?

2. In Sec 3.3, you probably mean $W_i \in R^{(im(f') + |s_j|) \times d}$. That is parenthesis are missing?

3. What are the +- std errors indicating? Is it cross validation error on a held-out test set?

4. Initially $i$ is indexing samples and by equation (3), (4) $i$ indexes time(?) and in Sec 3.3 $i$ indexes observations? How are observations defined here? is it measurement of specific modality at a specific time instance? Can you clear this in the introduction itself?

-----------------------------------------------------------------------------------------------------------------------------------------------------------------------
I have read the authors updated draft and response. The experiments section looks much better now.

1. The overall contribution has less clinical utility in my opinion as generally a patient likely deteriorates over time before an adverse outcome and therefore -- to give the model too much flexibility w.r.t. time ordering doesn't make quite as much sense. This is reflected in the fact that experimental results are not drastically better than other baselines. The authors might be able to show the utility of the method on other time series classification datasets where this is not a limitation of the data itself. However in those settings, it may be a bit hard to beat transformers. Do the authors have a sense of where the benefits of this method really are?

2. Mortality tasks are generally on the simpler side of clinical prediction problems as well. Nonetheless I think the contribution has some utility to the community. I do encourage the authors to try non--clinical datasets for a comparison

3. Please have a discussion that includes limitations and to discuss where the benefits of your methods really lie. A clear and thoughtful discussion is currently missing in your conclusions.

With that said, I am updating my score to a 6.

**Experience Assessment:**

I have read many papers in this area.

**Review Assessment: Checking Correctness Of Derivations And Theory:**

N/A

**Review Assessment: Checking Correctness Of Experiments:**

I assessed the sensibility of the experiments.

**Review Assessment: Thoroughness In Paper Reading:**

I read the paper at least twice and used my best judgement in assessing the paper.

---

> ### Author Response · Authors · 2019-11-11
> **Our comments to your review**
>
> > Decoupling “time encoding step” and “attention-based aggregation”
>
> Thanks for this suggestion! We added an ablation study to the appendix (Table A.4) and discuss it in the results section. Both factors (time encoding and attention) are shown to contribute to the overall performance. For time series with relatively comparable lengths (such as M3-Mortality), the time encoding improves performance only marginally, whereas for datasets with highly-varying lengths (such as M3-Phenotyping), time encoding leads to larger performance increases.
>
> > Key motivation for attention formulation
>
> The projection into the dot product space includes both the individual set elements and a global representation. It is thus possible to determine the importance of an individual observation using both global information about the whole time series, as well as the value and time point of this particular observation. The mechanism was designed this way to allow the model to have the possibility of selecting observations based on time, value, or modality alone. For example, the global representation could encode an average value of a channel so that individual observations would be attended to if they significantly deviate from this value.
>
> We have now extended and rewritten this section.
>
> > Is a_{j,i} shared across instances? [...]
>
> $a_{j,i}$ is calculated for *each* time series individually. We now clarify this in Section 3.3. Thanks for the suggestion!
>
> > It would be useful to provide how exactly a label is inferred for a *new* test instance.
>
> Thanks for the suggestion; we have updated Section 3.2 accordingly.
>
> Concerning your minor comments:
>
> We improved the introduction and added a citation for our claim.
> This  has been fixed, thank you very much!
> We clarify this now in the paper in Section 4.3.
> We changed the notation now in all the sections; for Section 3.3, we continue to use $i$ and $j$ for reasons of simplicity; we motivate their choice now.
>
> Please let us know if there are any other questions!

---

### Official Review · AnonReviewer2 · 2019-10-27
**Official Blind Review #2**

**Rating:** 6

**Review:**

This paper considers the problem of supervised classification of time-series data that are irregularly sampled and asynchronous, with a special focus on the healthcare applications in the experiments. Inspired by the recent progress on differentiable set function learning, the paper proposes an approach called Set Functions for Time Series (SEFT), which views the time series as sets, and use a parametrized sum-decomposing function f as the model for representing the probabilities of different classes, with the sets as the inputs. The problem then reduces to learning the finite dimensional parametrization of the function f under a given loss, which is a differentiable optimization problem that can be learned via standard optimization methods. Together with a positional embedding of the timestamps and an attention-based aggregation, the paper reports improved performance of the proposed approach on a few healthcare time series with asynchronous and irregularly sampled data. In particular, the runtime is largely shortened, while the final accuracy remains competitive to other methods compared in the paper.

The idea of SEFT is novel and the results are also showing its promise. In addition, the interpretability shown in section 4.3 is also attractive. However, there are several issues that limit the contribution and maturity of this paper.

Firstly, the paper proposes to model time series as a set. But this loses the information of the order of the time series, which can be extremely important in those datasets with long history dependence. In such cases, I'm not convinced that the set modeling would work. The authors should double check the characteristics of the datasets that are used, and see if they lack long history dependence properties in intuition. If so, this should be mentioned clearly. The authors should also make a more fair comparison with other approaches (like those based on RNN) on datasets with strong history dependence, e.g., Memetracker datasets of web postings and limit-order books datasets. Otherwise, it would be not clear whether this set modeling is generally applicable for general time series data.

Secondly, the authors missed a large amount of related literature for approaching asynchronous and irregularly sampled time series, namely (marked) point-process based approaches. See papers like [1, 2, 3], to name just a few. The authors should at least include some of the recent approaches in this direction for comparison before claiming the superiority of SEFT.

Thirdly, there are a few parts that are not very clear. 1) The discussion about complexity (order m and m\log m) at the bottom of page 1 is weird -- what does this complexity refer to? Does it include the learning of the unknown parameters in the models (like training of the neural networks in this paper)? 2) The loss function in formula (5) is not specified later in the paper (at least hard to find). 3) The Table 1 should be explained in much more details. In particular, why don't we include SEFT-ATTN for H-MNIST? The comment after * is also not clear to me -- is it relevant to why SEFT-ATTN is not included? And what are MICRO/MACRO/WEIGHTED AUC? And why are we using different sets of performance criteria for the first two and last two datasets?

Finally, some minor comments: 1) On page 2, "the following methods" should be "the above methods"; 2) on page 3, the meaning of "channels" should be specified clearer; 3) on page 4, in formulae (3) and (4), should there be \pi or 2\pi in the formula?

[1] Mei, Hongyuan, and Jason M. Eisner. "The neural hawkes process: A neurally self-modulating multivariate point process." Advances in Neural Information Processing Systems. 2017.
[2] Xiao, Shuai, et al. "Joint modeling of event sequence and time series with attentional twin recurrent neural networks." arXiv preprint arXiv:1703.08524 (2017).
[3] Yang, Yingxiang, et al. "Online learning for multivariate Hawkes processes." Advances in Neural Information Processing Systems. 2017.

############## post rebuttal ###############
After reading the authors' rebuttal, I decide to improve the rating to 5 (reflected as 6 due to the ICLR rating system limitation this year).


**Experience Assessment:**

I have published one or two papers in this area.

**Review Assessment: Checking Correctness Of Derivations And Theory:**

I carefully checked the derivations and theory.

**Review Assessment: Checking Correctness Of Experiments:**

I carefully checked the experiments.

**Review Assessment: Thoroughness In Paper Reading:**

I read the paper thoroughly.

---

> ### Author Response · Authors · 2019-11-11
> **Our comments to your review**
>
> >[...] But this loses the information [...]
>
> We understand this concern. However, we are convinced that no information about the ordering is lost. Since we encode the times into the set elements, it is completely possible for the model to account the ordering of time points in the time series. Nevertheless, we agree that (similar to the statement of R1), this approach does not contain a strong inductive bias for sequences. This would have stronger implications in forecasting or generative modelling scenarios, while in a pure classification setting, the indispensability of sequential orderings can be challenged (see “Order Matters”, Vinyals et al. 2016, where the authors show that RNNs achieve higher performance when processing sequences without their inherent ordering).
>
> > [...] long history dependence properties
>
> Our experiments did not specifically focus on long-term dependencies (we are convinced that medical time series could feature those dependencies, though). However, we would expect that our approach can exploit predictive long-term dependencies easier, because it has access to all information simultaneously. By contrast, RNNs have to propagate long-term dependencies through time, which remains challenging for very long time series. For this purpose, it would be interesting to run all methods on such time series with known long-term dependencies. We will tackle this large-scale problem in future work.
>
> > [...] Memetracker datasets of web postings and limit-order books datasets. [...]
>
> We want to emphasize that we focus on time series classification (TSC), and not time series modelling (forecasting, generative modelling etc.). Our empirically-supported claim is that SeFT is useful and scalable for TSC. Whether SeFT is meaningfully applicable to time series modelling (such as the meme tracker task) is an interesting question (to be considered in future work), albeit not essential to our current work.
>
> > Hawkes processes
>
> We now discuss Hawkes processes in the related work section of the paper. However, we would be grateful if the reviewer could point us towards literature that employs Hawkes processes in a classification scenario; to our knowledge, Hawkes processes are mostly used as generative models, and an extension to classification scenarios appears to be non-trivial.
>
> > The discussion about complexity (order m and m\log m) at the bottom of page 1 is weird -- what does this complexity refer to?
>
> We removed this sentence as it is indeed too confusing, especially as GP adapters and MGP adapters work with inducing points. Instead, we created a runtime plot in Figures A.1 and A.2 to support our scalability claims. Originally, we aimed to compare to GP adapter approaches. However, due to significant memory issues (on most tasks!) we were not able to include the GP adapter baseline (specifically, its multivariate extension MGP adapter Futoma et al. 2017).
>
> > The loss function in formula (5) is not specified later in the paper (at least hard to find)
>
> Thank you for pointing this out, it seems it was missed in the initial manuscript of the paper. For the multi-class classification we applied categorical cross-entropy loss where the final layer was set to utilize the softmax activation function. For multi-label and binary classification tasks sigmoid activation functions in combination with binary cross entropy loss were utilized.
> We updated the manuscript accordingly.
>
> > Table 1 details: Performance criteria for the first two and last two datasets?
>
> We require different metrics as the types of tasks are different. The first two datasets contain multi-class (H-MNIST) and multi-labels (Phenotyping) targets, whereas the last two datasets are binary classification tasks.
>
> > Table 1 details: MICRO, MACRO, WEIGHTED AUC
>
> For this, we follow the sklearn convention as found in ‘sklearn.metrics.roc_auc_score’:
> 'Micro': Calculate ROC metric globally by considering each element of the label indicator matrix as a label.
> 'Macro': Calculate ROC metric for each label, and find their unweighted mean. This does not take label imbalance into account.
> 'Weighted': Calculate ROC metric for each label, and find their average, weighted by support (the number of true instances for each label).
>
> We clarified these details in the caption of Table 1.
>
> > SEFT-ATTN for H-MNIST? [...]
>
> As H-MNIST merely includes 10 time steps, with missingness being randomly induced, we figured that the attention extension of our method is not meaningful for this specific setting. We added this clarification in the footnote of Table 1.
>
> > Minor comments
>
> Thanks for the suggestions! We fixed the typos and re-formulated as suggested and explained the meaning of channels. Moreover, concerning Eq. 3 and Eq. 4: we select our wavelengths between $2\pi$ and $\text{max\_ts} \cdot 2\pi$, so no additional $\pi$ parameter is necessary.
>
> Please let us know if there are any other questions!

---

### Author Response · Authors · 2019-11-11
**Thanks for your reviews**

We thank the reviewers for their thoughtful reviews. We have used them to update the paper and run several additional experiments. You may find the responses to your review as comments, containing further clarifications. Please let us know if there are any other questions.

---

### Decision · Program_Chairs · 2019-12-19

**Decision:**

Reject

**Comment:**

The paper investigates a new approach to classification of irregularly sampled and unaligned multi-modal time series via set function mapping. Experiment results on health care datasets are reported to demonstrate the effectiveness of the proposed approach.

The idea of extending set functions to address missing value in time series is interesting and novel. The paper does a good job at motivating the methods and describing the proposed solution. The authors did a good job at addressing the concerns of the reviewers.

During the discussion, some reviewers are still concerned about the empirical results, which do not match well with published results (even though the authors provided an explanation for it). In addition, the proposed method is only tested on the health care datasets, but the improvement is limited. Therefore it would be worthwhile investigating other time series datasets, and most important answering the important question in terms of what datasets/applications the proposed method works well.

The paper is one step away for being a strong publication. We hope the reviews can help improve the paper for a strong publication in the future.